# How Cells Deal with the Fluctuating Environment: Autophagy Regulation under Stress in Yeast and Mammalian Systems

**DOI:** 10.3390/antiox11020304

**Published:** 2022-02-02

**Authors:** Yuchen Lei, Yuxiang Huang, Xin Wen, Zhangyuan Yin, Zhihai Zhang, Daniel J. Klionsky

**Affiliations:** 1Life Sciences Institute, University of Michigan, Ann Arbor, MI 48109, USA; yclei@umich.edu (Y.L.); yxhuang@umich.edu (Y.H.); xinwen@umich.edu (X.W.); zyyin@umich.edu (Z.Y.); zhihaiz@umich.edu (Z.Z.); 2Department of Molecular, Cellular and Developmental Biology, University of Michigan, Ann Arbor, MI 48109, USA

**Keywords:** autophagy, energy stress, ER stress, nutrient stress, oxidative stress, regulation

## Abstract

Eukaryotic cells frequently experience fluctuations of the external and internal environments, such as changes in nutrient, energy and oxygen sources, and protein folding status, which, after reaching a particular threshold, become a type of stress. Cells develop several ways to deal with these various types of stress to maintain homeostasis and survival. Among the cellular survival mechanisms, autophagy is one of the most critical ways to mediate metabolic adaptation and clearance of damaged organelles. Autophagy is maintained at a basal level under normal growing conditions and gets stimulated by stress through different but connected mechanisms. In this review, we summarize the advances in understanding the autophagy regulation mechanisms under multiple types of stress including nutrient, energy, oxidative, and ER stress in both yeast and mammalian systems.

## 1. Overview of Autophagy in Yeast and Mammals

Autophagy is a highly regulated cellular degradation and recycling process, conserved from yeast to more complex eukaryotes [1]. The proteasome is responsible for degrading most short-lived, individual proteins, therefore, autophagy can degrade and recycle long-lived proteins, large protein complexes, and organelles [2]. The key definition of autophagy is the delivery and, typically, the degradation of cytoplasmic cargo within the lysosome (or the vacuole in fungi and plants) [3]. Based on different types of cargo and various modes of cargo delivery, at least three types of autophagy have been characterized, including microautophagy, macroautophagy, and chaperone-mediated auto-phagy/CMA; the latter process occurs in birds, fish, and mammals, but is not present in fungi [4]. The most comprehensively studied of these processes is macroautophagy, and, hereafter, we will use the term autophagy to refer to macroautophagy. The morphological hallmark of autophagy is the formation of the autophagosome, a large cytoplasmic double-membrane vesicle, which originates through the generation of the phagophore and the latter’s subsequent expansion and closure [5]. Once completed, the outer membrane of the autophagosome fuses with the lysosome/vacuole, while the inner membrane and cargo are exposed to the lumen of the degradative organelles for hydrolysis and the final efflux of the breakdown products into the cytosol [6].

The basic mechanism of autophagy has been well-documented, and the entire process of autophagy can be divided into the following stages: induction and nucleation of the phagophore, expansion and maturation of the phagophore into a completed auto-phagosome, docking and fusion with the lysosome/vacuole, and degradation and efflux of the breakdown products (Figure 1A) [7]. Initially identified in yeast, over 40 genes that have products primarily involved in the basic process of autophagy have been classified under the name autophagy-related (*ATG*) [8]. Many papers have provided details on this topic [2,4,9], therefore, here, we will briefly describe the autophagy process and the Atg proteins involved in both yeast and mammalian systems.

In yeast, the induction of autophagy begins at a single perivacuolar site, called the phagophore assembly site (PAS) which is proximal to the vacuole. This step is regulated by the Atg1 protein complex, including Atg1, Atg13, and the Atg17-Atg31-Atg29 ternary subcomplex [10,11]. In the nucleation stage, the Atg14-containing class III phosphatidylinositol (PtdIns) 3-kinase (PtdIns3K) complex I (consisting of Vps34, Vps30, Vps15, Atg14, and Atg38) is recruited to the PAS (Figure 1B) [12]. Next, the phagophore begins to expand and then seal to complete the formation of the autophagosome. Key components participating at this stage are two ubiquitin-like (Ubl) conjugation systems, which mediate the conjugation of Ubl proteins Atg12 and Atg8 [13,14] (Figure 1C). Through an enzymatic pathway involving Atg7 (an E1-like enzyme) for Atg12 activation and Atg10 (an E2-like enzyme) for Atg12–Atg5 conjugation, the C terminus of Atg12 is conjugated to an internal Lys of Atg5; Atg16 then noncovalently binds to Atg5 in the conjugate [15]. This system plays a role in membrane recruitment for the expanding phagophore. In contrast to Atg12, which is conjugated to another protein, Atg8 is conjugated to the lipid phosphatidylethanolamine (PE), allowing for its membrane association. Atg8 is initially synthesized with a C-terminal extension, which is removed by the Atg4 cysteine protease to expose a C-terminal Gly [16]. The modified Atg8 is activated with the help of Atg7, and then transferred to Atg3 (an E2 enzyme) that attaches the exposed C-terminal Gly to PE [13,15]. Atg8–PE is found on both sides of the phagophore and initially of the autophagosome; the portion on the autophagosome outer membrane will be deconjugated by a second Atg4-depedent cleavage when autophagosome formation is completed. The transmembrane protein Atg9 may cycle between the PAS and peripheral sites, thus carrying or directing the delivery of membrane for the expansion stage [17]. Upon maturation, the intact autophagosome fully surrounds the cargo, and ultimately delivers cargos to the vacuole by fusing with the vacuolar membrane. Finally, the cargo is degraded by various hydrolases in the vacuole, and breakdown products will be released back into the cytoplasm through permeases in the vacuole membrane.

There are some slight differences in components involved in the autophagy process in mammalian cells, whereas most components are homologs of Atg proteins in yeast. The activation of the ULK Ser/Thr kinase complex is required for autophagy induction, and the initiation begins with the ULK kinase complex (the catalytic subunits ULK1 or ULK2, the regulatory scaffold protein ATG13, RB1CC1, and the stabilizing protein ATG101) which can phosphorylate downstream factors for the induction of autophagy [9,18]. Next, the activated ULK1 complex phosphorylates and activates the PtdIns3K complex 1 (mainly composed of BECN1 [beclin 1], PIK3C3/VPS34, PIK3R4/VPS15, ATG14, NRBF2, and AMBRA1) [19]. The activated PIK3C3/VPS34 can phosphorylate PtdIns to produce phosphatidylinositol-3-phosphate (PtdIns3P), further contributing to formation of the phagophore [20]; ATG14 is directly associated with the ability of the PtdIns3K complex I to translocate to this site [21]. NRBF2 regulates PIK3C3 activity via promoting assembly of the complex, while AMBRA1 contributes to the interaction of BECN1 and PIK3C3 and the catalytic activity [22,23] (Figure 1B). Phagophores are nucleated on ER-emanating PtdIns3P-rich membrane domains called omegasomes [24]. In the expansion step, the ATG12 conjugation system (ATG12–ATG5-ATG16L1 complex) is like that in yeast (Figure 1C). In addition, the second Ubl system involves Atg8-family proteins including MAP1LC3/LC3 and GABARAP subfamilies, undergoing a similar process. With another protein, UVRAG, and core proteins from the PtdIns3K complex I, PtdIns3K complex II is formed, which is also important for autophagy. For example, UVRAG can bind to SH3GLB1 and promote autophagosome maturation [19].

Under normal conditions, autophagy keeps working constitutively at a basal state to maintain cellular homeostasis. When the cell is exposed to certain stress conditions, autophagy is massively induced and promotes the turnover of cytoplasmic materials required for cell survival or removing superfluous or damaged organelles. Too little or too much degradation from uncontrolled autophagy is harmful, and aberrant autophagy is associated with various diseases, such as cancer, aging, and neurodegeneration [25]. Autophagy can be either nonselective or selective: the nonselective mode degrades relatively random portions of the cytoplasm (although phase separation may be involved), whereas the selective mode is highly specific for certain components [26]. In particular, selective autophagy can degrade damaged and superfluous organelles or invasive microbes; these selective processes are given different names depending on the cargo, including mito-phagy (mitochondria), pexophagy (peroxisomes), aggrephagy (protein aggregates), lipophagy (lipid droplets), and xenophagy (intracellular pathogens) [26]. The selective mode of autophagy also plays a key role in cell physiology.

The frequently changing external environment causes cellular stress, and cells in a diseased state also experience stress from the unstable internal environment; therefore, stress biomarkers and their detection are important in the assessment of cell homeostasis (Table 1). To meditate metabolic adaptation and clear damaged organelles, autophagy is considered as one of the most important mechanisms to maintain cell survival under stress. Whereas the depletion of nutrients constitutes the main stimulus for massive autophagy induction, many other types of cellular stress are also involved in autophagy regulation. These different conditions can also regulate autophagy at different levels, including epigenetic, transcription, post-transcription, translation, and post-translation. In this review, we consider different stress stimuli and their relations with autophagy, with a goal of providing a more comprehensive understanding about this field.

## 2. Autophagy Regulation under Nutrient Stress

### 2.1. Mechanisms of Autophagy Regulation by Nutrient Stress in Yeast

Nutrients, such as amino acids and other nitrogen sources, are crucial for yeast growth and the target of rapamycin (TOR) pathway is the central regulator [52]. Tor1 and Tor2 are conserved protein kinases that can be found in two protein complexes termed TOR complex 1 (TORC1) and TORC2. TORC1, which consists of Tor1 or Tor2, Kog1, Lst8, and Tco89, is particularly sensitive to rapamycin treatment and activated by nutrients [52]. Unlike TORC1, TORC2 can only utilize Tor2 as its catalytic component and is not sensitive to rapamycin [53]. Even though some studies indicate that TORC2 is involved in promoting autophagy [54], TORC1 is still considered as the master regulator of autophagy, especially under nutrient stress.

Nitrogen and amino acid signaling are transmitted to TORC1 via different mechanisms, which largely involve the conserved RAG GTPase, composed of Gtr1 and Gtr2 [55]. Gtr1 and Gtr2 form a heterodimer and are tethered on the vacuole membrane by the Ego1-Ego2-Ego3/EGO complex [52]. In the presence of sufficient nitrogen sources/amino acids, with the help of Vam6 as a guanine nucleotide exchange factor/GEF, Gtr1 and Gtr2 are in the GTP and GDP bound forms, respectively. Activated Gtr1 binds to Kog1 and Tco89 in the TORC1 complex, trapping and activating TORC1 on the vacuole [56]. When the amino acid level goes down, Seh1-associated subcomplex inhibiting TORC1/SEACIT, which is composed of Npr2, Npr3, and the catalytic subunit Iml1, functions as the GTPase activating protein (GAP) to induce GDP loading onto Gtr1; this form no longer activates TORC1 [57].

TORC1 regulation function with regard to autophagy can be summarized in three aspects (Figure 2A). First, TORC1 directly phosphorylates Atg13 under growing conditions, which prevents Atg13-Atg1 complex activity [58,59]. The inactivation of TORC1 by nutrient deprivation leads to the hypophosphorylation of Atg13, the induction of Atg1 kinase activity and autophagy stimulation [58].

Second, TORC1 regulates the transcription of *ATG* genes. TORC1 negatively regulates the expression of genes required for the adaptation to nutrient stress by regulating the expression and localization of several transcription factors (TFs), including Gcn4, and the GATA-binding proteins Gln3 and Gat1 [60,61]. When TORC1 is inactivated by the depletion of nutrients, the activated phosphatases Sit4 and PP2A (Pph21/22-Tpd3-Cdc55) mediate the translocation of Gln3 and Gat1 to the nucleus [60], where they are required for the successful induction of *ATG7, ATG8, ATG9, ATG29,* and *ATG32* expression [62]; Gln3 is necessary for promoting *ATG14* expression [63]. TORC1 inhibition also induces Gcn4 expression [61] and *GCN4* deletion leads to decreased *ATG1* mRNA level and autophagy activity during starvation [62]. Additionally, Gcn4 is responsible for the increased expression of *ATG41* during starvation, which is required for efficient autophagy [64]. In addition to directly regulating the localization of TFs, TORC1 and one of its downstream targets, Sch9, inhibit the translocation of Rim15 from the cytosol to the nucleus; Rim15 is a kinase, which controls the association between several TFs and *ATG* genes [27,65]. For instance, Rph1 is a transcriptional repressor of several *ATG* genes, including *ATG7*, *ATG8*, *ATG9*, *ATG14,* and *ATG29* under nutrient-rich conditions. Upon nutrient stress, Rph1 is phosphorylated by the nuclear-localized Rim15 and dissociates from the *ATG* genes to induce autophagy [66]. Similarly, Ume6 inhibits *ATG8* transcription when nutrients are abundant, and this inhibition is relieved by Rim15-dependent phosphorylation during starvation [67].

Third, TORC1 controls the posttranscriptional regulation of *ATG* genes. In a report by Hu et al., a temperature sensitive mutation in the decapping enzyme Dcp2 results in increasing mRNA level of multiple *ATG* genes under nutrient-replete conditions, including *ATG1, ATG8, ATG9,* and *ATG13*, and a higher autophagy activity [68]. In the same study, the researchers also found that in *C. neoformans*, TORC1 phosphorylates Dcp2 under nutrient-rich condition, thus promoting *ATG8* mRNA degradation [68].

Here, we only summarized autophagy regulation related to TORC1. Multiple layers of regulation happen in both growing and/or starvation conditions to regulate autophagy as listed in Table 2, but their relations to nutrient stress signals are not completely elucidated.

Although additional factors have been identified that regulate autophagy in yeast during starvation, many questions still remain. First, some autophagy regulators play dual roles under growing and starvation conditions (Table 2). For example, Dhh1 contributes to the degradation of multiple *ATG* mRNAs under nutrient-rich conditions, but, on the contrary, promotes the translation of *ATG* genes during starvation [68,73,81]. However, how this transition happens and how it connects with a nutrient-sensing pathway is still unclear. Second, TORC1 has different localizations based on nutrient status. When nutrients are replete, TORC1 is activated and disperses along the vacuole membrane; several studies indicate that TORC1 forms punctate structure on the vacuole in response to starvation [82,83,84]. However, it is still unclear whether the change in TORC1 localization contributes to autophagy regulation. One model proposes that because the PAS is formed close to the vacuole, the dispersed TORC1 localization along the vacuole prevents Atg13 recruitment to the PAS. Conversely, TORC1 puncta formation during starvation limits its access to Atg13, providing more opportunities for hypophosphorylated Atg13 to be recruited to the PAS [82]. A recent study indicates that the EGO complex and TORC1 have two localizations, both on endosomes and the vacuole but only the TORC1 on endosomes controls autophagy through targeting Atg13 [85]. However, it is not clear why TORC1 has these two different pools and, considering that the distribution between them does not change during nitrogen starvation, why these two populations of TORC1 function differently. Third, epigenetic regulation is another critical way to control autophagy at an appropriate level. Although not much is known in yeast, some evidence indicates the relationship between TORC1, histone modifications, and autophagy regulation. A study from Füllgrabe et al. identified that autophagy occurs concomitant with the reduction in histone H4 Lys16 acetylation/H4K16ac, which may result from the autophagic degradation of the acetyltransferase Sas2 [80]. More recently, Set2, a histone methyltransferase, was shown to be necessary for the transcriptional response to nutrient stress; Set2 genetically interacts with Tor1 and Tor2, indicating a potential role in autophagy regulation [86], but further studies are needed to reveal the mechanism of this type of regulation.

### 2.2. Autophagy Regulation in Mammalian Cells

In mammalian cells, nutrient starvation is also a common stress that induces auto-phagy. Multiple important nutrient-response molecules have been reported to regulate autophagy, among which MTOR (mechanistic target of rapamycin kinase) complex 1 (MTORC1) is the best characterized. In this subsection, we will summarize autophagy regulation mechanisms mediated by MTORC1 and briefly introduce some other molecules that contribute to autophagy regulation under nutrient stress.

#### 2.2.1. Autophagy Regulation by MTORC1

Like in yeast, there are two TOR complexes in mammalian cells, MTORC1 and MTORC2. MTORC1, which consists of MTOR, RPTOR/raptor, DEPTOR, LST8, and PRAS40, is the general responder to growth factors and nutrients [87]. Amino acids are essential for the activation of MTORC1 through RRAG GTPases [88,89]. Mammalian cells contain four RRAG GTPase members, RRAGA, RRAGB, RRAGC, and RRAGD and they functions in a heterodimer, in which one monomer of either RRAGA or RRAGB partners with either RRAGC or RRAGD [90]. Amino acids in the lysosomal lumen activate the Ragulator complex, possibly through the vacuolar-type H^+^-translocating ATPase (V-ATPase) [91,92], and the Ragulator complex functions as a guanine nucleotide exchange factor that promotes the active conformation of RRAG GTPase; where RRAGA/B binds with GTP and RRAGC/D is loaded with GDP [91,92,93]. In addition, amino acids inhibit the GATOR1 complex (analogous to yeast SEACIT), the GAP for RRAGA/B, therefore facilitating the activation of the RRAG GTPase [94]. Once activated, RRAG heterodimer binds with RPTOR and brings MTORC1 into proximity with RHEB (Ras homolog, mTORC1 binding) GTPase on lysosomes [88]. MTORC1 activity is coupled with growth factors; the removal of TSC1-TSC2, the GAP of RHEB GTPase, in response to growth factors, allows the activation of MTORC1 by RHEB [95]. In contrast, the lack of nutrients results in the conversion of RRAG GTPase into its inactive form and the lysosomal localization of TSC2, which inhibits RHEB GTPase activity [96]. In addition, the absence of amino acids inhibits the polyubiquitination of RHEB, which is important for its binding with MTORC1 [97]. As a result, MTORC1 becomes inactivated and displays a cytosolic localization.

MTORC1 is considered as the master regulator of autophagy when cells are facing nutrient stress. Autophagy regulation by MTORC1 can be summarized in the following aspects (Figure 2B): First, MTORC1 regulates the posttranslational modification of autophagy-associated proteins. Several ATG proteins are the direct targets of MTORC1 and the best known are ULK1 and ATG13. MTORC1-dependent phosphorylation of ULK1 and ATG13 reduces ULK1 complex activity. During starvation, the inactivated MTORC1 disassociates from ULK1, relieving inhibition of the latter; subsequent triggering of ULK1 complex activity promotes autophagy [98,99,100]. In return, activated ULK1 inhibits MTORC1 activity via phosphorylating RPTOR and reducing its substrate-binding ability [101,102]. This feedback loop maintains the inactivation of MTORC1 and is important for the full activation of autophagy during nutrient deprivation.

Several components of the PtdIns3K complex are also targets of MTORC1. In complex I, MTORC1-dependent phosphorylation of ATG14 inhibits the kinase activity of the complex [103]. Another component, NRBF2, can be phosphorylated by MTORC1 at Ser113 and Ser120. MTORC1 inhibition suppresses NRBF2 phosphorylation and changes it binding preference from PIK3C3/VPS34 and PIK3R4/VPS15 to ATG14 and BECN1, supporting PtdIns3K complex I assembly and its association with the ULK1 complex [104]. In addition, AMBRA1 is phosphorylated by MTORC1 at Ser52 and becomes inactivated. Upon nutrient stress, activated AMBRA1 interacts with the E3 ligase TRAF6 and ubiquitinates ULK1, enhancing its activity [105]. In complex II, UVRAG is the direct target of MTORC1 [106]. MTORC1-dependent phosphorylation on Ser498 has a positive effect on the interaction between UVRAG and RUBCN, which negatively regulates PIK3C3/VPS34 kinase activity and the interaction between HOPS (a complex involved in tethering) and UVRAG, therefore inhibiting the fusion between autophagosomes and lysosomes [106]. Apart from the proteins in these two complexes, WIPI2 can be phosphorylated by MTORC1 at Ser395, which promotes its polyubiquitination by HUWE1 and subsequent degradation [107].

Besides directly phosphorylating ATG proteins, MTORC1-dependent phosphorylation of the acetyltransferase EP300 prevents its intra-molecular inhibition, thus activating its catalytic activity [108]. Several ATG proteins, including ATG5, ATG7, ATG8, and ATG12 are the targets of EP300, and the acetylation of these ATG proteins inhibits autophagy [109]. Therefore, under nutrient stress, MTORC1 inactivation reduces the acetylation of essential ATG proteins to fully activate autophagy.

Second, MTORC1 controls the transcription of *ATG* genes and lysosomal genes via regulating the localization of several TFs. TFEB is a member of the basic helix-loop-helix leucine-zipper family of TFs that promotes the transcription of genes in lysosomal biogenesis and autophagy [110]. When nutrients are repleted, TFEB will be recruited to the lysosome by active RRAG GTPase and phosphorylated by MTORC1 [111], whereas starvation leads to a rapid translocation of TFEB from the cytosol to the nucleus and the induction of transcription of autophagy-associated genes such as *UVRAG, WIPI, MAPLC3B, SQSTM1, VPS11, VPS18,* and *ATG9B* [112]. Several MTORC1-dependent phosphorylation sites are found on TFEB, including Ser122, Ser138, Ser142, and Ser211 [113,114,115,116], which regulate TFEB cellular localization through different but coordinated mechanisms. Phosphorylation on Ser211 by MTORC1 promotes TFEB association with YWHA/14-3-3 (tyrosine 3-monooxygenase/tryptophan 5-monooxygenase activation protein) proteins, which traps TFEB in the cytosol. Inactivation of MTORC1 leads to the transport of TFEB to the nucleus, thus stimulating the transcription of autophagy-associated genes [113,114]. Ser122 is another MTORC1-dependent phosphorylation site. The phosphorylation mimetic mutation Ser122Asp reduces nuclear TFEB when Ser211 is dephosphorylated but Ser122 dephosphorylation is not sufficient, in itself, to result in the nuclear localization of TFEB [115], indicating that Ser122 coordinates with Ser211 to control TFEB localization. Ser138 and Ser142 are localized in proximity to the nuclear export signal/NES on TFEB, and MTORC1-dependent phosphorylation at these two sites is critical for TFEB nuclear export [116]. Besides the direct regulation of TFEB, MTORC1 inhibits TFEB activity through activating KAT2B/GCN5, an acetyltransferase that acetylates TFEB and inhibits its DNA-binding activity [117]. Therefore, under nutrient stress, MTORC1 inhibition stimulates not only TFEB accumulation in the nucleus but its binding to target genes as well, thus promoting lysosomal biogenesis and autophagy flux.

TFE3 and MITF are additional TFs that drive the expression of genes involved in lysosome biogenesis and autophagy [118,119,120]. Similar to TFEB, TFE3 and MITF are recruited to lysosomes by activated RRAG GTPase, and MTORC1 inactivation during starvation is necessary for their release from YWHA/14-3-3 proteins and nuclear localization [111,118]. Ser321 on TFE3 is apparently an MTORC1-dependent phosphorylation site because both MTORC1 inactivation and Ser321Ala mutation abolish its interaction with YWHA/14-3-3 proteins and stimulate nuclear localization [118]. On MITF, Ser280, a residue that corresponds to TFEB Ser211 by homology analysis, is also nominated as a potential MTORC1 phosphorylation site [111]. Interestingly, a study found that TFEB and TFE3 positively regulate MTORC1 activity by promoting RRAGD expression and recruiting MTORC1 to the lysosome when nutrients are provided to the starved cells. Even though it remains as an open question as to how modulating RRAGD expression is sufficient for regulating MTORC1 activity, this mechanism may be important for cells to prepare for nutrient refeeding during starvation [121].

In addition to directly phosphorylating TFs and affecting their localization, MTORC1 also regulates the EIF2A (eukaryotic translation initiation factor 2A)-ATF4 pathway, which induces the expression of *ATG* genes [122,123]. During nutrient deprivation, MTORC1 inactivation induces PPP6C (protein phosphatase 6 catalytic subunit) phosphatase activity, which dephosphorylates and activates EIF2AK4/GCN2. EIF2AK4/GCN2 further phosphorylates and activates EIF2A, leading to the subsequent increase of ATF4 expression, *ATG* gene transcription induction and activated autophagy [122].

Third, MTORC1 is responsible for the posttranscriptional regulation of *ATG* genes. Recently, MTORC1 is reported to regulate autophagy via controlling mRNA N^6^-methyl-adenosine (m^6^A) methylation. In this study, the researchers found that MTORC1 activates the CCT (chaperonin containing TCP1) complex, which helps in the folding of proteins in the m^6^A methyltransferase complex, resulting in more m^6^A RNA methylation, the degradation of *ATG* transcripts and the suppression of autophagy [124].

Finally, in recent years epigenetics has been proposed to be an important regulatory aspect of autophagy [125]. Even though not much is known about the regulation of histone modifications by MTORC1 during nutrient stress, some studies provide initial clues. One example is that during starvation or rapamycin treatment, acetylation of histone H4 Lys16/H4K16ac and the corresponding acetyltransferase KAT8/hMOF are both downregulated, which is important for cell survival during starvation although the detailed mechanism is not known [80]. In addition, MTORC1 enhances the nuclear localization of FOXK1 and FOXK2, which recruit the SIN3A-HDAC complex to restrict the acetylation of histones and the expression of *ATG* genes [126]. These two examples suggest a close connection between MTORC1 and the epigenetic regulation of autophagy, but more studies are still needed to better understand this relationship.

#### 2.2.2. Other Autophagy Regulation during Nutrient Stress

In addition to MTORC1, the main sensor of nutrients, several other stress-response kinases regulate autophagy during nutrient stress. For instance, the stress-activated signaling molecule MAPK8/JNK1 phosphorylates BCL2 during starvation, which prevents its interaction with BECN1, thus promoting autophagy [127]. MAPKAPK2 and MAPKAPK3, which belong to the stress-response kinase MAPK family, phosphorylates BECN1 Ser90 during starvation and this phosphorylation is important for BECN1 function [128]. Additionally, the IKK complex gets activated by starvation and induces the expression of several autophagy genes, therefore stimulating autophagy [129]. As mentioned above, multiple stress-response kinases contribute to the stimulation of autophagy during nutrient starvation, but whether, and how, these signaling pathways coordinate to regulate autophagy requires further attention.

## 3. Autophagy Regulation under Energy Stress

Recycling by autophagy is essential for yeast and mammals to survive starvation. The breakdown products and materials can be further used to provide building blocks for the synthesis of essential proteins and to produce ATP through catabolic pathways. Therefore, autophagy is essential for the maintenance of energy homeostasis and is finely regulated upon energy deprivation.

AMP-activated protein kinase (AMPK) is an evolutionarily conserved serine/threonine protein kinase [130], sensing low cellular ATP levels and controlling turnover of cellular materials and metabolism, thus being essential for cellular adaptation to energy limitation [131]. AMPK is a heterotrimeric complex composed of a catalytic subunit (PRKAA/α) and two regulatory subunits (PRKAB/β and PRKAG/γ). The PRKAA-subunit contains the kinase domain and the critical residue Thr172 whose phosphorylation by upstream kinases activates AMPK activity [132]. The PRKAG-subunit possesses four cystathionine β-synthase/CBS motifs that can bind to all forms of adenosine-containing ligands, enabling it to sense the changes in the ATP:AMP/ADP ratio [133,134], and the activity of AMPK is precisely regulated by these ratios in the cell [130]. When cells are in the fed state, AMPK is mostly bound by ATP and its activity is inhibited. Under energy-starvation conditions, the cellular concentration of ATP decreases whereas levels of ADP and AMP increase. AMP or ADP binding to the PRKAG-subunit activates the kinase through three distinct mechanisms: (1) it promotes the STK11/LKB1 (serine/threonine kinase 11)-mediated phosphorylation of the PRKAA subunit at Thr172, which can increase AMPK activity up to 100-fold in vitro [135,136]; (2) it protects phosphorylated Thr172 from dephosphorylation by phosphatases [137]; and (3) it causes allosteric activation of the AMPK complex [138]. Once activated, AMPK serves as a central metabolic regulator to restore energy homeostasis by inhibiting anabolic pathways and promoting catabolic pathways, including autophagy. AMPK promotes autophagy at various steps by phosphorylating autophagy-related proteins or autophagy regulators.

Activated AMPK can induce the autophagic process by inhibiting the activity of MTOR in two ways (Figure 3): (1) AMPK directly phosphorylates the MTORC1 component RPTOR on Ser722 and Ser792. This phosphorylation induces YWHA/14-3-3 binding to RPTOR, thus hindering the binding of RPTOR to MTOR and MTOR substrates, leading to suppression of MTORC1 activity [139]. (2) AMPK phosphorylates the MTOR upstream regulator TSC2 on Thr1227 and Ser1345, which promotes the GTPase-activating function of the TSC1-TSC2 complex, leading to the transformation of RHEB into an inactive RHEB-GDP state, which consequently reduces MTOR activity [140,141]. As mentioned above, reduced MTOR activity relieves the inhibition on ULK1 to activate autophagy.

AMPK can also stimulate autophagy through phosphorylating autophagy related proteins including ULK1, BECN1, and PIK3C3/VPS34 (Figure 3). Under energy-starvation conditions, AMPK directly activates ULK1 through phosphorylation of Ser317, Ser467, Ser555, Ser574, Ser637, and Ser777 [98]. This activation is prevented by MTOR activity during normal physiological conditions as MTORC1 phosphorylates ULK1 at Ser757, which is located in the AMPK-ULK1 binding region (amino acids 711-828), thereby inhibiting the interaction between AMPK and ULK1 [98]. AMPK also regulates the PIK3C3/VPS34 lipid kinase complex upon glucose withdrawal: AMPK activates the pro-autophagy PIK3C3/VPS34 complex by phosphorylating Ser91 and Ser94 in BECN1, which increases autophagosome formation. In the meantime, AMPK inhibits the PIK3C3/VPS34 complexes not involved in autophagy by phosphorylating Thr163 and Ser165 in PIK3C3/VPS34 to suppresses overall PtdIns3P production. The presence of ATG14 dictates the differential regulation by inhibiting PIK3C3/VPS34 phosphorylation and increasing BECN1 phosphorylation by AMPK during glucose starvation [142]. Furthermore, AMPK can phosphorylate other core components of the autophagy pathway. For example, activated AMPK can phosphorylate ATG9A at Ser761, which recruits ATG9A to LC3-positive autophagosomes and enhances autophagosome production [143].

Apart from direct phosphorylation of the core components of the autophagy machinery, AMPK can also promote autophagy through activating autophagy regulators (Figure 3). For example, human transcription factor FOXO3 is phosphorylated by AMPK at Thr179, Ser399, Ser413, Ser555, Ser588, and Ser626, which promotes the nuclear translocation of FOXO3 and its activity, thus upregulating the transcription of downstream auto-phagy-related genes such as *ATG4*, *ATG12*, *BECN1*, *LC3*, and *ULK1* [144,145]. The NAD-dependent deacetylase SIRT1 (sirtuin 1), an essential regulator of autophagy during energy deprivation, is also under the regulation of AMPK. Under glucose starvation conditions, cytoplasmic GAPDH is phosphorylated by activated AMPK and redistributes into the nucleus, where it interacts with SIRT1 and displaces SIRT1′s repressor CCAR2/DBC1 leading to the activation of SIRT1 [146]. SIRT1 can also be activated by the increased level of NAD^+^ during starvation. The targets of SIRT1 include, but are not limited to, autophagy pathway components ULK1, ATG5, and LC3 and the transcription factor FOXO1, which induces the expression of the GTPase RAB7 that mediates the fusion of autophagosomes with lysosomes [80,147,148]. During energy stress, a considerable amount of AMPK is translocated from the cytosol to mitochondrial-associated ER membrane/MAM, where it interacts with and phosphorylates the mitochondrial fusion protein MFN2. This AMPK-MFN2 axis is required for mitochondrial-associated ER membrane dynamics and auto-phagy induction [149].

## 4. Autophagy Regulation under Oxidative/Nitrosative Stress

Reactive oxygen and nitrogen species (hereafter ROS and RNS) are highly reactive molecules that can cause oxidative damages on macromolecules and biological membranes [150,151]. Cells have developed very sophisticated mechanisms to regulate the homeostasis of ROS and RNS, including endogenous antioxidants, such as glutathione and TXN (thioredoxin), and detoxifying enzymes, such as GPX (glutathione peroxidase), CAT (catalase), and SOD (superoxide dismutase), to efficiently resolve the excessive oxidative stress [151,152]. In coordination with the ubiquitin–proteasome system, autophagy plays essential roles in sequestering oxidized proteins in the lysosome/vacuole for degradation to maintain homeostasis [153,154,155]. However, autophagy is also involved in oxidative stress-induced cell death [156]. For example, the free iron released by ferritinophagy could promote lipid ROS accumulation, thus triggering ferroptosis and the increased autophagic flux in SOD1^G93A^ transgenic lead to muscle atrophy [157,158,159]. To date, there is abundant evidence showing that the autophagy activity is tightly regulated by the oxidative stress [154,160,161,162].

As for the ROS, the direct reaction between oxygen and extra electron gives rise to the superoxide (O_2_^●−^), which is highly reactive and is rapidly converted into hydrogen peroxide by the endogenous SOD [163]. H_2_O_2_ is relatively stable and is considered as an important signaling molecule for the ROS responsive pathways [164,165]. In the presence of iron, H_2_O_2_ can generate the unstable hydroxyl radical (HO^●^) via a process called the Fenton reaction. Hydroxyl radicals can further react with polyunsaturated fatty acid to form various lipid peroxides [166,167]. Autophagy is activated by each of these different ROS species as well as numerous RNS species [156,168,169,170,171,172,173,174,175,176].

In this section, we cover the current understanding of the regulatory mechanisms of autophagy in yeast and mammalian cells. The relationship between mitophagy and oxidative stress has been reviewed by De Gaetano et al., in the same special issue [177].

### 4.1. Mechanisms of Autophagy Regulation by Oxidative Stress in Yeast

The regulatory role of oxidative stress on autophagy is evolutionarily conserved in the budding yeast *Saccharomyces cerevisiae*. For example, yeast mitophagy induced by nitrogen starvation and ethanol challenge both can be prevented by adding the antioxidant N-acetylcysteine/NAC [178,179].

Yap1 signaling is the most well-characterized oxidative stress responsive pathway in yeast [180,181]. Yap encompasses a transcription factor family of eight basic leucine zipper (bZIP) domain proteins [182,183]. Among them, Yap1 can directly translocate into the nucleus to activate the expression of various antioxidant genes such as *TRX2* by the stimulation of oxidative stress [184,185]. Under basal conditions, Yap1 is enriched in the cytosol as the nuclear exporter Crm1 efficiently pumps Yap1 out of the nucleus [185]. Upon H_2_O_2_ activation, however, Yap1 is oxidized, and several disulfide bonds are formed on its C-terminal cysteine-rich domain/c-CRD and amino-terminal cysteine-rich domain/n-CRD so that the Crm1-cognate nuclear export signal is masked. As a result, Yap1 is trapped in the nucleus where it activates the expression of stress-responsive genes [186,187]. The oxidation of Yap1 (especially the covalent bonds between Cys303 and Cys598) requires the participation of the thiol peroxidase Hyr1/Gpx3/Orp1, which acts as a direct receptor for H_2_O_2_ [188]. Yap1 recognizes a consensus DNA element in the promoter region called the Yap response element (YRE, which includes TGACTAA, TTAGTCA, TTACTAA, and T[T/G]ACAAA) [180]. Among all currently known *ATG* genes, only *ATG15* contains a potential binding site for Yap1. *ATG15* encodes a vacuolar phospholipase that can break down the inner autophagosome membrane in the vacuole lumen, and its direct activation by Yap1 has been experimentally verified (Figure 4A) [189,190].

Atg4 is a cysteine protease, and the mammalian homolog has been reported to be directly regulated by H_2_O_2_ (see below for further details) [191]. However, the Cys81 residue on human ATG4A and ATG4B that is proposed to play important roles in this process is not conserved in yeast. Yeast Atg4 is also redox regulated through a different mechanism: site-directed mutagenesis reveals that a single disulfide bond formed by Cys338 and Cys394 has a very low redox potential and is required for Atg4 redox regulation in yeast; the formation of this disulfide bond decreases the Atg4 protease activity and can be rapidly reduced by thioredoxin [192].

### 4.2. Mechanisms of Autophagy Regulation by Oxidative Stress in Mammalian Cells

In mammalian cells, ROS accumulation can be triggered by several different stimuli, such as hypoxia, nutrient stress or cytokines including TNF/TNFα [193,194,195,196,197]. Therefore, many upstream signaling pathways have been proposed to affect the activity of auto-phagy, including signaling by NFKB/NF-κB, AMPK, HIF1A/HIF-1, ATM, AKT-MTOR, and MAPK [194,195,197,198,199,200,201,202,203,204]. The regulation of autophagy by these upstream signaling pathways has been well summarized elsewhere [153,154,160,205].

The NFE2L2/Nrf2 (NFE2 like2 bZIP transcription factor 2)-KEAP1 (kelch ECH associated protein 1)-antioxidant signaling pathway can be directly activated by oxidative stress and can stimulate the expression of various stress-responsive genes including several detoxifying enzymes and autophagy proteins [206,207,208]. As a functional ortholog for Yap1, NFE2L2 is also a bZIP transcription factor; heterodimers of NFE2L2 and MAF proteins recognize a specific antioxidant response element/ARE in the promoter region of target genes [209,210]. Under basal conditions, the NFE2L2 is localized in the cytoplasm and maintained at a very low level. This dynamic regulation is achieved by its interacting partner KEAP1 which is a CUL3 (cullin 3) E3 ubiquitin ligase adaptor [211,212,213]. KEAP1 interacts with NFE2L2 via the carboxy-terminal Kelch domain with a 2:1 stoichiometry. At the same time, KEAP1 interacts with the CUL3 ligase via the amino-terminal bric-a-brac, tramtrack, and broad complex/BTB domain, thus promoting the efficient degradation of NFE2L2 [211,214,215,216]. KEAP1 acts as the redox sensor and the interaction between KEAP1 and NFE2L2 is directly regulated by environmental cues via a mechanism called the “hinge and latch model”: in response to H_2_O_2_, KEAP1 Cys226, Cys613, and Cys622/624 residues form disulfide bonds that impair the interaction between KEAP1-CUL3 and NFE2L2, thus stabilizing NFE2L2 and releasing it into the nucleus where it is active [212,213,217,218,219].

The SQSTM1/p62 (sequestosome 1) protein contains an LC3-interacting region (LIR) domain thus allows it to act as an autophagy receptor to facilitate delivery of cargos into the phagophore for subsequent degradation [220,221]. The antioxidant response element has been identified in the promoter of *SQSTM1* that makes it a target for NFE2L2 activation [222]. Interestingly, KEAP1 is among the autophagic substrates of SQSTM1. Therefore, SQSTM1-mediated autophagy can degrade KEAP1 to further activate NFE2L2 signaling in a positive feedback loop [223,224,225,226]. Recently, more NFE2L2-targeted autophagy genes have been reported, including *ULK1*, *CALCOCO2*, *ATG4D*, *ATG7*, *GABARAPL1*, *ATG2B*, *ATG5*, and *LAMP2B*, suggesting that, in addition to TFEB and FOXO, NFE2L2 is an important autophagy regulator, perhaps in a more oxidative-stress relevant context (Figure 4B) [227,228].

In addition, ROS can regulate autophagy by directly oxidizing the cysteine residues on the core autophagy components. For example, the cysteine protease ATG4A and ATG4B can be inactivated by H_2_O_2._ The possible mechanism is that Cys81 is sensitive to oxidation, triggering a conformation change on Cys77 which is the catalytic residue, thus inhibiting the cysteine protease activity of ATG4 [191]. As a result, the transient blockade of ATG4 activity stabilizes the lipidated forms of its substrates LC3 and GABARAPL2/GATE-16 so that autophagosome biogenesis is promoted [191]. Of note, this reversible inhibition of ATG4 is spatially and temporally regulated. As the autophagosome is trafficked towards lysosomes where the local H_2_O_2_ is lower, ATG4 is reactivated to deconjugate and recycle the LC3 and GABARAPL2. Similarly, in the context of mito-phagy, the cysteine residues of the ubiquitin E3 ligase PRKN/parkin can be oxidized by sulfhydration that is required for full PRKN ligase activity and normal mitophagy flux [229,230].

### 4.3. Mechanisms of Autophagy Regulation by Nitrosative Stress

In both yeast and mammals, nitric oxide (NO) mediates critical physiological functions as a signaling molecule at low concentrations, but causes nitrosative stress at high concentrations [231,232,233]. Imbalance of reactive nitrogen species (RNS) results in accumulation of protein tyrosine nitration, protein S-nitrosylation on cysteine residues, and damage to lipids and DNA [234,235]. In mammalian systems, nitrosative stress is correlated with many pathological conditions, such as cancer, neurodegenerative diseases, and ischemia, and upregulated autophagy activity is observed in several nitrosative stress models [236,237,238]. For example, in a microsphere embolism rat model, the increased autophagy signaling (protein level of BECN1, LC3, LAMP2, and CTSB [cathepsin B]) is accompanied by nitrosative stress, which can be partially resolved by adding the peroxynitrite (ONOO^−^) scavenger melatonin [239]. Furthermore, RNS can attenuate MTORC1 activity to promote autophagy via the ATM-AMPK-TSC2 and AKT signaling axis [240,241]. A recent study challenged MCF7 cells with the NO donor compound DETA-NONOate and observed an increased NAD^+^:NADH ratio. The pharmacological and genetic inhibition of the NAD^+^-dependent deacetylase SIRT1 reduces autophagy activity, and the acetylation of TP53, and promotes cell survival, suggesting the complex interplay among SIRT1, TP53, and autophagy upon nitrosative stress (Figure 4C) [242]. However, whether autophagy is required for RNS homeostasis will requires further loss-of-function studies of autophagy genes and their products.

## 5. Autophagy Regulation under ER Stress

### 5.1. Endoplasmic Reticulum Stress and Autophagy

The endoplasmic reticulum (ER) is a central membrane-bound organelle, and its membrane structure was first documented by Porter et al. in 1945 using electron microscopy [243]. The ER is an important organelle in eukaryotic cells with various functions, such as protein synthesis, modification and processing of proteins, secretion of correctly folded proteins, calcium homeostasis, and lipid and carbohydrate metabolism [244,245]. Therefore, the ER is essential for cell homeostasis.

Normally, the ER utilizes chaperones to properly fold newly synthesized proteins and identify misfolded proteins for destruction. However, the ER homeostasis is disrupted under numerous pathological conditions including nutrient deprivation, perturbation of cellular ATP level, calcium metabolic imbalance, redox imbalance, viral infection, and the presence of environmental toxins. In addition, the protein-folding capacity of the ER can also be compromised and eventually cause the accumulation of unfolded or misfolded proteins in the ER lumen, also known as ER stress [246].

The ER stress triggers an adaptive response referred to as the unfolded protein response (UPR). After sensing the ER stress, the UPR transduces the signal to the regulation of downstream transcription factors and then induces ER chaperone genes to upregulate the folding capacity. Additionally, the cells can also begin a process termed ER-associated degradation (ERAD) to mediate the transport of unfolded or misfolded proteins into the cytosol for degradation. ERAD mainly consists of two mechanisms: ubiquitin–proteasome-dependent ERAD/ERAD(I) and autophagy–lysosome dependent ERAD/ERAD(II) [247,248,249]. ER-to-lysosome-associated degradation/ERLAD is the name currently used for the autophagy-dependent mechanism that is employed to handle proteins that cannot be degraded by ERAD.

As mentioned in the previous section in this review, autophagy can be induced by different types of cellular stress, which includes the ER stress discussed here. The relationship between ER stress and autophagy was first described in 2006 in yeast [250,251]. Here, we name the autophagy activated by ER stress as “ER stress-mediated autophagy” because this is the term used in most studies.

### 5.2. The Mechanisms of ER Stress-Mediated Autophagy in Yeast

In yeast, the ER stress can be sensed by an ER-resident type 1 transmembrane protein called Ire1, which plays a critical role in the UPR induced by ER stress [252,253]. It is noteworthy that Ire1 was initially identified as an mRNA splicing factor in yeast [254,255]. In addition, Ire1 is also capable of sensing unfolded or misfolded proteins in the ER lumen because Ire1 has both an endoribonuclease domain and an ER lumenal stress-sensing domain.

Ire1 is localized in the ER membrane with its C terminus facing into the cytosol and the N terminus residing in the ER lumen. Under normal conditions, the N-terminal region of Ire1 is bound to Kar2 unless Ire1 senses the accumulation of unfolded proteins in the ER lumen. Ire1 is activated by autophosphorylation after dissociation from Kar2, leading to the expression of activated Hac1, a transcription factor.

Upon the UPR induced by ER stress, a non-classical intronic sequence near the 3’ end of the open reading frame of *HAC1* mRNA is excised by activated Ire1 and then the two ends of the mRNA are ligated by the tRNA ligase Trl1 [256]. The spliced *HAC1* encodes an activated form of the Hac1 protein containing 238 amino acids, which contains 18 amino acids more than the Hac1 protein encoded by un-spliced *HAC1* mRNA; these 18 amino acids play a key role for TF activation [257]. The difference in the properties of the two types of Hac1 protein is mainly caused by the C terminus. The N terminus of both types of Hac1 has a DNA-binding function while the C terminus of activated Hac1 has an active transcriptional activation domain due to the cleavage and splicing reaction [258]. Eventually, activated Hac1 is exported to the nucleus and binds to the unfolded protein response elements/UPREs to promote the transcription of UPR-related genes [259]. Furthermore, studies have reported that these unfolded protein response elements are commonly found in the promoters of some UPR-related genes including *FPR2/FKB2*, *KAR2,* and *PDI1* [260].

Studies carried out in 2006 show that ER stress can induce autophagy through this Ire1-Hac1 pathway in yeast [250,251]. Yorimitsu et al. used two types of drugs: dithiothreitol/DTT (an inhibitor of disulfide bond formation) and tunicamycin (an inhibitor of glycosylation) to induce ER stress. Then GFP-Atg8 processing and precursor Ape1 maturation assays were applied to monitor autophagic induction after the drug treatment. Both assays showed increased autophagic flux, indicating an induction of autophagy caused by ER stress. Additionally, the necessity of the Ire1-Hac1 signaling pathway for this ER stress-mediated autophagy was also explored in this study. The authors found that deletion of either *IRE1* or *HAC1* does not affect the capability for inducing autophagy caused by nutrient depletion; however, either knockout did block ER stress-mediated autophagy, suggesting that Ire1 and Hac1 are involved in the induction of this pathway probably through the UPR.

### 5.3. The Mechanisms of ER Stress-Mediated Autophagy in Mammalian Cells

The UPR is a highly conserved mechanism and mammalian cells also utilize it to alleviate ER stress by enhancing the protein-folding capacity of, and reducing the protein synthetic load on this organelle to restore ER homeostasis [259]. Unlike the UPR in yeast, that consists of the Ire1 signaling pathway, the UPR in mammalian cells is characterized by three major branches involving three ER membrane resident proteins: the serine/threonine-protein kinase/endoribonuclease ERN1/IRE1α (endoplasmic reticulum to nucleus signaling 1), EIF2AK3/PERK (eukaryotic translation initiation factor 2 alpha kinase 3), and the cyclic AMP-dependent transcription factor ATF6 (activating transcription factor 6).

Under normal physiological conditions, all three ER stress sensors are inactive due to the binding of an ER-resident chaperone protein, HSPA5/BIP/GRP78. Due to the high affinity of unfolded or misfolded proteins for HSPA5, ERN1, EIF2AK3, and ATF6 become active when there is an accumulation of unfolded or misfolded proteins in the ER lumen [261]. Moreover, the increased activity of these three ER sensors is also partially contributed to by unfolded or misfolded proteins acting as active ligands for their activation [262]. The activation of these three UPR signaling pathways alleviates ER stress by partially overlapping but distinct mechanisms, including autophagy.

#### 5.3.1. ERN1

ERN1 is a bifunctional protein in mammalian cells consisting of three domains: an N-terminal lumenal domain, a cytosolic endoribonuclease domain, and a cytosolic serine/threonine kinase domain [263]. Similar to the Ire1 in yeast, active ERN1 can excise a 26-nucletide intron from *XBP1* mRNA. The spliced *XBP1* mRNA allows the expression of an active and stable form of XBP1 (X-box binding protein 1). The transcription factor XBP1 is then translocated to the nucleus and upregulates the expression of target genes in response to ER stress [264]. Among the target genes, *BECN1* plays a central role in autophagy, suggesting that the splicing of *XBP1* mRNA mediated by ERN1 under ER stress is important for autophagy induction [265]. Consistently, studies report that the un-spliced *XBP1* mRNA can interact with FOXO1 (forkhead box O1), resulting in a decreased level of this TF, finally leading to the downregulation of autophagy [266,267,268].

Additionally, ERN1 can interact with TRAF2 (TNF receptor associated factor 2) and form a complex, which can phosphorylate MAP3K5/ASK1 (mitogen-activated protein kinase kinase kinase 5). Next, the phosphorylated MAP3K5 actives MAPK8/JNK1 (mitogen-activated protein kinase 8) by phosphorylation. Subsequently, the phosphorylated MAPK8-mediated phosphorylation of BCL2 can increase the level of free BECN1 by disrupting the BECN1-BCL2 complex, or by elevating *BECN1* transcription, which leads to autophagosome formation [127,269]. Moreover, it is reported that the activation of AMPK mediated by ERN1 is involved in autophagy initiation [270].

#### 5.3.2. EIF2AK3

Under ER stress conditions, the activation of the EIF2AK3 UPR signaling pathway upregulates many autophagy-related genes. The active EIF2AK3 can mediate the phosphorylation of EIF2A, which can elevate both *ATG12* mRNA and protein levels [271]. In addition, EIF2AK3-mediated EIF2A phosphorylation also enables the selective translation of *ATF4* mRNA, and the transcription factor ATF4 is then translocated to the nucleus where it upregulates the expression of multiple proteins, such as several autophagy-related proteins (ATG3, ATG12, ATG16L1, BECN1, and LC3) and DDIT3/CHOP (DNA damage inducible transcript 3) [272]. The expression of DDIT3 can also transcriptionally increase the expression of some proteins involved in autophagy (ATG5, ATG10, and GABARAP). In addition, DDIT3 can downregulate the expression of BCL2, a protein that binds to BECN1 and inhibits autophagosome formation [273,274]. Interestingly, the complex formed by ATF4 and DDIT3 can also induce the expression of some proteins involved in autophagy, including ATG7, NBR1 (NBR1 autophagy cargo receptor), and SQSTM1 [123]. Furthermore, the active EIF2AK3 pathway can initiate autophagy via the activation of AMPK and the inhibition of MTORC1 [275]. Consistent with this finding, the activation of ATF4-DDIT3 mediated by EIF2AK3 inhibits MTORC1 activity resulting in the induction of autophagy [276].

#### 5.3.3. ATF6

Under ER stress conditions, ATF6 is translocated to the Golgi apparatus where it is cleaved by MBTPS1/S1P and MBTPS2/S2P. The N-terminal domain of ATF6 after the cleavage is translocated to the nucleus to induce the expression of UPR genes, including *DDIT3* and *XBP1* [277,278,279]. Therefore, ATF6 can indirectly regulate autophagy through the DDIT3 and XBP1 signaling pathway as mentioned above. In addition, ATF6 might regulate autophagy in the initiation step by the inhibition of AKT activity [280]. Additionally, ATF6 can interact with the transcription factor CEBPB (CCAAT enhancer binding protein beta) and then stimulate the expression of DAPK1 (death associated protein kinase 1) [281,282], which can phosphorylate BECN1 so that it will be released from the auto-phagy inhibitory BECN1-BCL2 complex, promoting the induction of autophagy.

#### 5.3.4. Calcium

The ER is a multifunctional organelle which plays a pivotal role in maintaining intracellular calcium homeostasis. Under ER stress conditions, the calcium homeostasis is disrupted and the release of calcium from the ER to the cytosol is also elevated, which can induce autophagy. When calcium is released from the ER through ITPR (inositol 1,4,5-trisphosphate receptor), the CAMKK-AMPK-dependent signaling pathway is activated and the inhibitory effect of MTOR on the ULK1 complex is relieved [283,284]. Moreover, calcium release can activate DAPK1 [285], which is involved in the induction step of autophagy as noted above.

## 6. Conclusions

In this review, we summarized autophagy regulation under different types of stress, including that involving nutrients, energy, oxidation, and the ER (Figure 5). Besides the ones mentioned in this review, other types of stress, such as DNA damage and pathogen infection, are also able to induce autophagy [286,287]. The fact that autophagy is induced by multiple stresses highlights the importance of autophagy in allowing cells to maintain homeostasis in response to changes in the environment.

Of note, in this review, we mainly focus on how stress-response molecules and/or pathways regulate autophagy. There are a wide range of regulatory mechanisms affecting autophagy-associated genes under stress conditions, especially during nutrient deprivation, but the connections with stress-sensing pathways have not been established [288]. Additionally, apart from stress, the activity of stress-sensing molecules can also be regulated by other factors, and all these contribute to autophagy regulation under stress conditions. We did not discuss these factors in detail because they are beyond the scope of this review, but their roles in autophagy regulation cannot be ignored.

Even though we introduced different types of stress separately, it does not mean that the stress-responding molecules or pathways function alone. In fact, stress-responding pathways have very close connections, regulating each other or sharing the same downstream effector. Using the nutrient sensor MTORC1 and energy sensor AMPK as an example, it is well known that AMPK inhibits MTORC1 and both kinases target ULK1 to regulate its activity [53,98]. The regulatory network among AMPK, ULK1, and MTORC1 are important for the oscillation of autophagy [289]. Recently, AMPK was shown to be inhibited by MTORC1 [290], further indicating the complex interaction between these two important stress responders. Another example is seen with EIF4A, which is activated by multiple stresses and induces the expression level of ATF4, a transcription factor that promotes the transcription of multiple *ATG* genes [272]. Similarly, NFE2L2 is also in the center of the stress response, as the expression of this transcription factor is activated not only by oxidative stress, but also by other conditions such as ER stress [291]. The interaction between different stress-responding pathways and the existence of a common response pathway makes autophagy induction by stress a rapid and well-controlled process.

Thanks to advanced studies in recent decades, we can now draw a clearer picture of the autophagy regulation network under stress conditions. However, more studies focusing on this field are still needed for a better understanding on how autophagy is controlled, because too much or too little autophagy can harm cells. More importantly, the insights on autophagy regulation under stress may shed light on understanding the relation between autophagy and disease because cells in a diseased state usually undergo stress. For example, cancer cells in the interior of a tumor usually experience nutrient and oxidative stress because of the lack of proximal blood vessels. ER stress and oxidative stress are also proposed to contribute to neurodegenerative diseases [292,293]. Autophagy has a close connection with these diseases [18], therefore, a deeper understanding of autophagy regulation under stress conditions may help us find more potential autophagy-targeting therapeutic approaches.

## Figures and Tables

**Figure 1 antioxidants-11-00304-f001:**
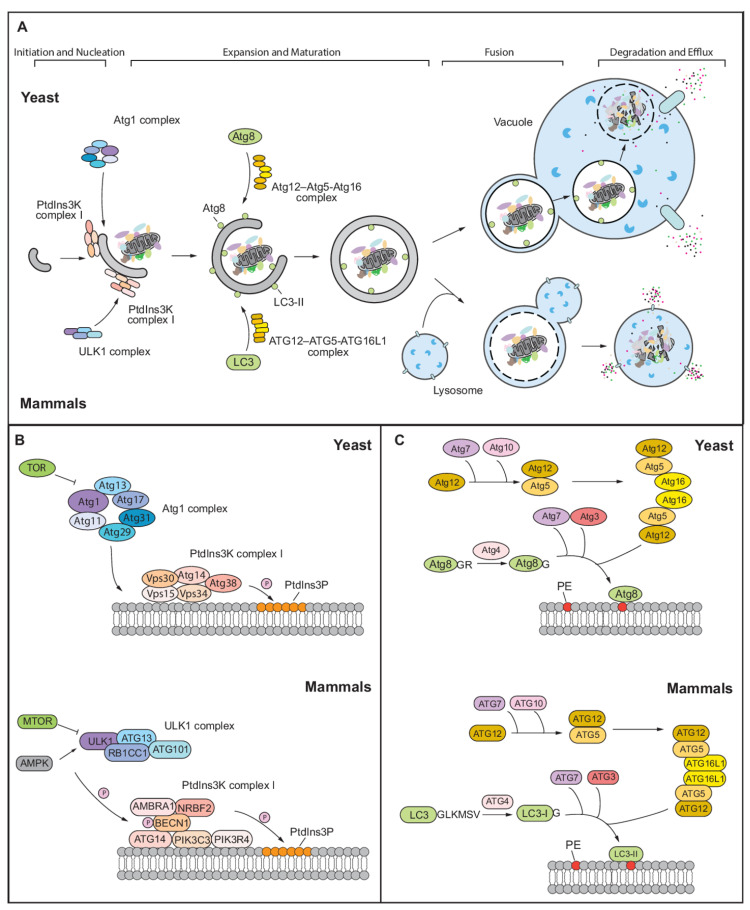
Autophagy in yeast and mammalian systems. (**A**) Four stages of autophagy. The upper and lower parts of each panel represent yeast and mammals, respectively. In both yeast and mammals, autophagy includes four stages, induction and nucleation of the phagophore, expansion and maturation of the phagophore, fusion with the vacuole (in yeast)/lysosome (in mammals), and degradation and efflux of the breakdown products. (**B**) Protein complexes involved in induction and nucleation of the phagophore. In yeast, Atg1 and PtdIns3K complex I will be recruited to the PAS and drive the formation of PtdIns3P on the phagophore. In the mammalian system, the ULK1 complex phosphorylates and activates PtdIns3K complex I, which contributes to the formation of the phagophore. (**C**) Two ubiquitin-like systems. In both yeast and mammals, the Atg12 complex (Atg12–Atg5-Atg16 in yeast and ATG12–ATG5-ATG16L1 in mammals) forms with the help of Atg7/ATG7 and Atg10/ATG10; this complex then functions as an E3 enzyme for the conjugation of Atg8 (in yeast) and Atg8-family proteins (LC3 and GABARAP subfamilies in mammals) with PE.

**Figure 2 antioxidants-11-00304-f002:**
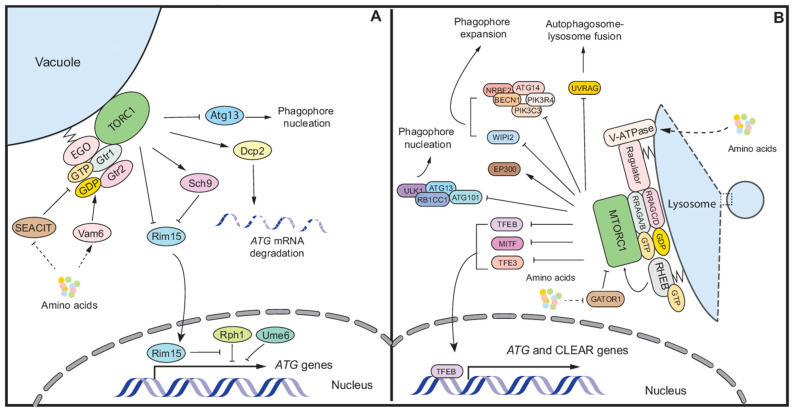
Autophagy regulation by TORC1 and MTORC1. (**A**) In yeast, nutrient sources activate TORC1 basically through activated Gtr1-Gtr2 heterodimer. The activated MTORC1 inhibits auto-phagy through phosphorylating Atg13, inducing degradation of *ATG* mRNA and inhibiting Rim15 translocation to nucleus, which releases some repressing transcription factors such as Rph1 and Ume6 and induces *ATG* genes transcription. (**B**) In mammalian cells, amino acids activate RRAG GTPase and recruit MTORC1 to lysosomes, where it is activated by RHEB. Activated MTORC1 phosphorylates several ATG proteins and inhibits their functions. Meanwhile, MTORC1 dependent phosphorylation in TFEB, MITF, and TFE3 blocks their translocation into nucleus and the induction of *ATG* and CLEAR genes transcription. Solid and dashed lines represent direct and indirect regulations respectively.

**Figure 3 antioxidants-11-00304-f003:**
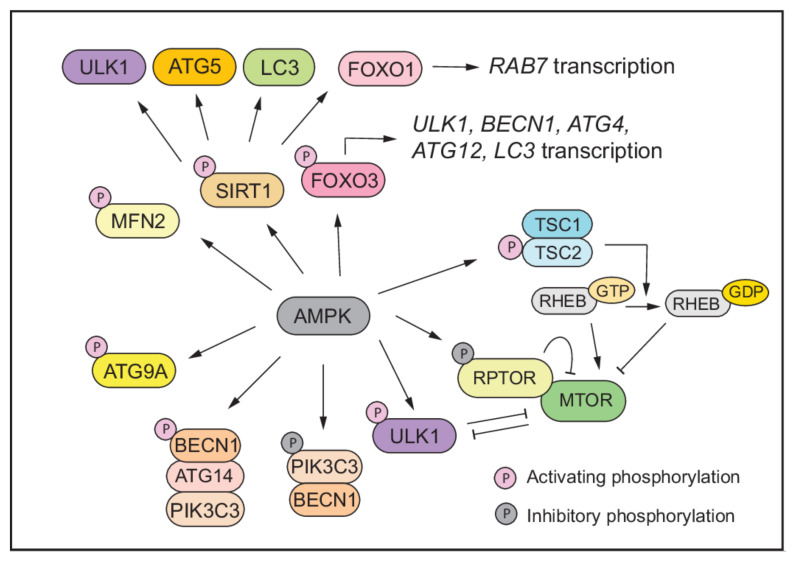
AMPK drives autophagy through three layers of regulation. (1) AMPK suppresses MTOR activity by phosphorylating TSC2 and RPTOR. (2) AMPK directly phosphorylates and activates proteins involved in autophagy including ULK1, BECN1, PIK3C3/VPS34, and ATG9A. (3) AMPK activates the positive regulators of autophagy, for example, phosphorylation of FOXO3 leads to increased transcription of autophagy-related genes.

**Figure 4 antioxidants-11-00304-f004:**
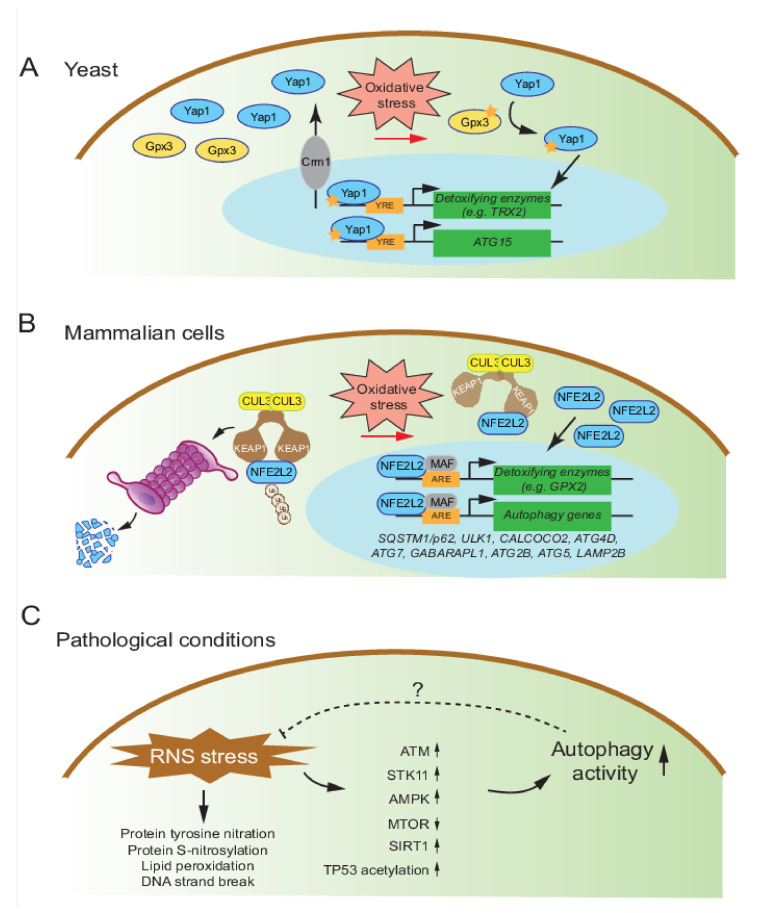
The transcriptional regulation of autophagy by oxidative stress. (**A**) In yeast, upon oxidative stress, oxidized Yap1 is accumulated in the nucleus and drives the expression of detoxifying enzymes and the autophagy-related gene *ATG15*. YRE: Yap response element. (**B**) In mammalian cells, upon oxidative stress, the oxidation of cysteine residues of KEAP1 prevent the ubiquitination of NFE2L2/NRF2, thus allowing NFE2L2 to enter the nucleus and activate several autophagy genes. ARE: antioxidant response element. (**C**) In pathological conditions, nitrosative stress activates autophagy activity via multiple signaling cascades.

**Figure 5 antioxidants-11-00304-f005:**
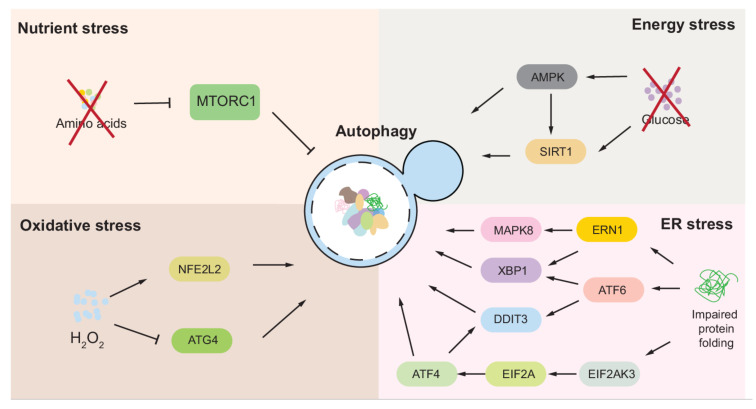
Summary of autophagy regulation under stress. Mammalian systems are presented here to demonstrate the major causes of stress, the main stress-sensing molecules and pathways and their regulation of autophagy.

**Table 1 antioxidants-11-00304-t001:** Stress biomarkers and their detection.

Stress Type	Organism	Biomarkers	Detection	Reference
Nutrient stress	Yeast	TORC1 inactivation	Sch9 dephosphorylation	[27]
			Mammalian EIF4EBP1 dephosphorylation in vitro	[28]
	Mammals	MTORC1 inactivation	RPS6KB1 dephosphorylation	[29]
			EIF4EBP1 dephosphorylation	[29]
Energy stress	Yeast and mammals	Lower ATP: ADP/AMP ratio	Liquid chromatography to detect ATP, ADP and AMP level	[30]
			ATP:ADP fluorescence reporter	[31,32]
			Bioluminescent detection	[33]
	Yeast	Snf1 activation	“SAMS” peptide phosphorylation	[34]
	Mammals	AMPK activation	AMPK phosphorylation	[35]
			Phosphorylation of downstream targets such as ACAC (acetyl-CoA carboxylase)	[35]
Oxidative stress	Yeast and Mammals	High level of ROS	Dichlorodihydrofluorescein fluorescence	[36]
			PG1 or PC1 fluorescence	[37]
			Calcein-acetoxymethylester (calcein-AM) fluorescence	[38]
			CellROX dye	[39]
		Increased GSSG:GSH ratio	High-performance liquid chromatography	[40]
			Capillary electrophoresis	[40]
			Bioluminescence	[41]
			Genetically-encoded fluorescent sensors	[42,43]
		Lipid peroxidation	Fluorescence shift of C11-BODIPY (581/591)	[44]
			TBA-MDA assay	[45,46]
ER stress	Yeast	Misfolded protein accumulation	Kar2 sedimentation	[47]
		UPR pathway activation	Transcription reporter containing a UPR element promoter driving fluorescent proteins	[47]
			Ire1 clustering	[47]
	Mammals	Protein aggregates	Thioflavin T (ThT) fluorescence	[48]
		UPR pathway activation	Spliced *XBP1* mRNA detection using ER stress-activated indicator” (ERAI) construct	[49]
			Upregulated expression of UPR target genes, including *DDIT3* and *HSPA5/GRP78*	[50]
			ATF6 translocation	[51]

**Table 2 antioxidants-11-00304-t002:** Autophagy regulation in yeast under nutrient-rich and starvation conditions.

Type of Regulation		Regulatory Factors	Conditions	Effects (↑, Positive; ↓, Negative)	Target Genes or Proteins	Reference
Transcriptional regulation		Pho23	Nutrient-rich	↓	*ATG1, 7, 8, 9, 12, 14, 29*	[69]
		Spt10	Nutrient-rich	↓	*ATG1, 7, 9,14, 32*	[62]
		Fyv5	Nutrient-rich and starvation	↓	*ATG1, 7, 8, 9,14, 29, 32*	
		Sfl1	Nutrient-rich	↓	*ATG1, 7, 8, 9, 14, 29, 32*	
		Sko1	Nutrient-rich and starvation	↓	*ATG1, 7, 8, 32*	
		Zap1	Nutrient-rich	↓	*ATG1, 7, 8, 9, 14, 29, 32*	
		Swi5	Nutrient-rich	↑	*ATG7, 8, 9, 14, 29*	
		Rsc1	Starvation	↑	*ATG8*	[70]
		Spt4/5	Nutrient-rich	↓	*ATG8, 41*	[71]
			Starvation	↑	*ATG41*	
Post-transcriptional regulation		Xrn1	Nutrient-rich	↓	*ATG1, 4, 5, 7, 8, 12, 14, 16, 29, 31*	[72]
		Dhh1	Nutrient-rich	↓	*ATG3, 7, 8, 19, 20, 22, 24*	[68]
			Starvation	↑	*ATG1,13*	[73]
		Pat1	Starvation	↑	*ATG1, 2,7, 9*	[74]
		Psp2	Starvation	↑	*ATG1, 13*	[75]
		Ded1	Starvation	↑	*ATG1*	[76]
Post- translational regulation	Phosphorylation	Hrr25	Cvt pathway and pexophagy induction	↑	Atg19,36	[77]
	Ubiquitination	Met30	Nutrient-rich	↓	Atg9	[78]
	Acetylation	Esa1	Starvation	↑	Atg3	[79]
	Deacetylation	Rpd3	Starvation	↓	Atg3	[79]
Epigenetic regulation	Acetylation	Sas2	Nutrient-rich	↓	Histone H4 Lys16	[80]
	Methylation	Unclear	Nutrient-rich	↓	Histone H3 Lys4	[80]

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
