# Peer review of "How Cells Deal with the Fluctuating Environment: Autophagy Regulation under Stress in Yeast and Mammalian Systems"

_antioxidants, 2022, doi:10.3390/antiox11020304_

Round 1

Reviewer 1 Report

The manuscript "How cells deal with the fluctuating environment: Autophagy 2 regulation under stress" is a reviewing article.
For a reviewing article, some issue should be included.

1. Figure 2 show the transcriptional regulation of autophagy by oxidative stress, authors could also show the regulation of autophagy by nitrosative stress.
2. Please also list a table to show the stress associated biomarkres.
3. Authors may also list a table to show how to detect the stress associated biomarkers.

Reviewer 2 Report

Dear colleagues! 

As a peer reviewer assigned by the Editor I have the following summary on the manuscript by Lei et al. submitted to Antioxidants.

Generally, the Review article is well-written and provides a clear read to the audience with a focus on molecular effectors that regulate autophagy under different stressful conditions that disturb the cell's homeostatic patterns. The paper is written and proofed thoroughly so only minor typos/style errors were found and will be listed below.

Nevertheless, I suggest the Issue Editors to draw their own opinion on compliance of the Review's focus with the Special issue thematics as far at the paper's descriptive part goes far beyond the RSS/ROS/RNS yet I feel that it is a valuable contribution from a strong group eminent in the field.

After careful review of the manuscript I have the following queries and remarks:

- Title: majority of comparative logics of the paper is by drawing Reader's attention to yeast mechanisms followed by mammalian cells' regulation. I feel that may be this might be highlighted in the Title by mentioning yeast and mammalian cells. 

1) Overview of Autophagy in Yeast and Mammals is an important section yet I feel that a minimal schematics (showing most crucial regulators) is required as far as for a Reader new to the field lines 48-95 are extremely hard to comprehend and summarize just by reading the Text. 

2) In Figure 1 Rph1 repressive role is not obvious from the A panel bottom part. Such imagung more relates to promoting role rather than repressing, so I suggest to add a stub arrow to highlight its function.

3) I also feel that in all figures translocation by a factor should be marked by a specific arrow type to discern between interaction and spatial movement of a molecule 

4) Table 1 - probably "-" and "+" might by highlighted by arrows "up" and "down" as far as "-" is often used to fill an empty a cell in table (that lacks content) so in current view the table is easily skipped by a Reader while it has important data. in Table 1 Column 2 must be widened to avoid word breakdown "Acetylati-on, Deacetyl-ation etc." in its bottom lines.

5) In Section 2.1 role and functions of TORC2 are omitted, but I suggest to add a sentence explaining this choice and at least providing a short knowledge on TORC2 to the Reader.

6) in line 225 "Ragulator complex appears yet it is not explained that it includes the particiaption of RRAGx members mentioned above so it may puzzle a Reader new to the field.

7) In Fig 2B association between MTORC1 and ULK1 mentioned in lines 245-246 is not obvious so I suggest to harmonize these parts

8) Section 3 would benefit from a schematic figure to demonstrate regulatory patterns involved in AMPK control and its role in autophagy.

9) In line 441 "numerous RNS species" are mentioned while not a single example of RNS radical is provided in the section

10) Figure 2A - YAP1* translocation is not demonstrated so the regulatory cascade seems incompletely shown

11) Ref-ce to Figure 3 appear only in line 676, but it definitely should be put earlier in the text in corresponding section.

12) In figure 3 central part (formation of autophagosome and its fusion) must be named clearly indicating the organelle and process

Minor/typo issues:

Line 152 "of nutrient" - replace by "to nutrient" ?

Line 219 "like yeast" - replace by "like in yeast"

Line 224 - "functions in a geterodimer" - replace by "function in heterodimers"

Line 276 - typo "MOTRC1"

Line 314 - typo "PPP6" for 'PP6" and second time "phosphatase" after brackets is obsolete

Line 320 - typo "on mRNAs" might be replaced by "in mRNAs"

Line 340 - typo "phosphorylates" might be replaced by "phosphorylate"

Lines 430 and 432 - sentence seems misleading and missing words or conjugation between parts.

Line 432 - "there are abundant" should be replaced by "there is abundant"

Line 478 - "redox regulated" might be replaced by "redox-regulated"

Line 486 - "stimulate" should be replaced by "stimulates"

Lines 491-495 - the sentence is too ling and hard for understanding so I suggest to break it in 2 phrases

Lines 525-530 are more appropriate at the start of section rather that as a concluding part

Line 602 - comma after "on, this organelle" is obsolete

Author Response

  1. “Title: majority of comparative logics of the paper is by drawing Reader's attention to yeast mechanisms followed by mammalian cells' regulation. I feel that may be this might be highlighted in the Title by mentioning yeast and mammalian cells.”
    We agree with the reviewer. We have added ‘in yeast and mammalian systems’ in the title as suggested.

  2. “Overview of Autophagy in Yeast and Mammals is an important section yet I feel that a minimal schematics (showing most crucial regulators) is required as far as for a Reader new to the field lines 48- 95 are extremely hard to comprehend and summarize just by reading the Text.”
    We think this is a very valuable suggestion. We have added a schematic figure of autophagy (Fig. 1 in the revised manuscript) and we think that it will help with the understanding of the overall autophagic process.

  3. “In Figure 1 Rph1 repressive role is not obvious from the A panel bottom part. Such imagung more relates to promoting role rather than repressing, so I suggest to add a stub arrow to highlight its function.”
    We appreciate this suggestion. We have added a stub arrow/inhibitory bar to show that Rph1 and Ume6 inhibit the transcription of ATG genes (now Fig. 2). At the same time, we added a similar inhibitory bar for nuclear-localized Rim15 to indicate that it helps in the release of these transcription factors and induces autophagy.

  4. “I also think that in all figures translocation by a factor should be marked by a specific arrow type to discern between interaction and spatial movement of a molecule.”
    This is an interesting suggestion, and we agree. We have changed the type of arrow showing the spatial movement to distinguish them from the arrows indicating induction.

  5. “Table 1 - probably "-" and "+" might by highlighted by arrows "up" and "down" as far as "-" is often used to fill an empty a cell in table (that lacks content) so in current view the table is easily skipped by a Reader while it has important data. in Table 1 Column 2 must be widened to avoid word breakdown"Acetylati-on, Deacetyl-ation etc." in its bottom lines.”
    Again, this is a very constructive comment. We have changed the table (now Table 2) ‘-’ and ‘+’ to ‘ ¯’ and ‘­’ to make it clearer. We also widened column 2 to avoid the word separation.

  6. “In Section 2.1 role and functions of TORC2 are omitted, but I suggest to add a sentence explaining this choice and at least providing a short knowledge on TORC2 to the Reader.”
    This is a good idea. We have added a brief introduction to TORC2 and the reason why we focus on TORC1 in the revised manuscript (lines 169-171 in the manuscript with tracked changes).
  7. “In line 225 "Ragulator complex appears yet it is not explained that it includes the particiaption of RRAGx members mentioned above so it may puzzle a Reader new to the field.”

In the sentence being referred to (lines 332-346 in the revised manuscript), we mentioned the Ragulator complex to explain how RRAG members are activated by amino acids because the Ragulator works as the guanine nucleotide exchange factor (GEF) of RRAG. In addition, in lines 338-340, we mentioned that activation of RRAG brings MTORC1 to the lysosome where it gets activated. We think these explanations for how RRAG members participate in the activation of MTORC1 in response to amino acids and their relationship with the Ragulator complex will provide a sufficient explanation.

  1. “In Fig 2B association between MTORC1 and ULK1 mentioned in lines 245-246 is not obvious so I suggest to harmonize these parts.”

We think that this is a very good suggestion; however, in Figure 2B, because we would like to show how MTORC1 regulates autophagy as comprehensively as we can, it is hard to show how MTORC1 controls ULK1 activity specifically. However, the point that MTORC1 inhibits ULK1 complex activity is shown in the figure.

  1. “Section 3 would benefit from a schematic figure to demonstrate regulatory patterns involved in AMPK control and its role in autophagy.”

We think that this is another helpful suggestion. We have added a new figure to demonstrate how AMPK regulates autophagy (Fig. 3 in the revised manuscript).

  1. “In line 441 "numerous RNS species" are mentioned while not a single example of RNS radical is provided in the section.”

The reviewer made a valid point, which is similar to the first comment of Reviewer #1. As stated above, we have added a new section (section 4.3) and a new panel (Fig. 4C) to demonstrate how RNS regulates autophagy.

  1. “Figure 2A - YAP1* translocation is not demonstrated so the regulatory cascade seems incompletely shown.”

We have modified this figure (now Figure 4) so that YAP1* translocation is shown.

  1. “Ref-ce to Figure 3 appear only in line 676, but it definitely should be put earlier in the text in corresponding section.”

Former Figure 3 (now Figure 5) is the figure summarizing the four types of stress we mentioned in the main text. This figure only shows the most important stress-responding molecules and how they regulate autophagy, and therefore, we only refer this figure in the Conclusion. To address point #9, we added a schematic figure in section 3, so we have a more detailed figure showing regulatory mechanism in each type of stress that we discussed except for ER stress. We think that the figures now clearly show how autophagy is regulated under stress.

  1. “In figure 3 central part (formation of autophagosome and its fusion) must be named clearly indicating the organelle and process.”

In this figure (now Figure 5) we added a label to the middle as “autophagy” because the purpose of this figure is to summarize how autophagy is regulated under different types of stress but not to show the autophagic process per se. The central part, which is the autophagosome and its fusion with a lysosome, is the hallmark of autophagy and we want to use it to represent the autophagic process. To address point #2, we added a schematic figure showing the entire process of autophagy and we think with that figure, the central part of Fig. 5 in the revised manuscript will make more sense.

  1. “Minor/typo issues:

Line 152 "of nutrient" - replace by "to nutrient" ?

Line 219 "like yeast" - replace by "like in yeast"

Line 224 - "functions in a geterodimer" - replace by "function in heterodimers"

Line 276 - typo "MOTRC1"”

We appreciate the reviewer pointing out these errors. We have fixed the above four typos.

  1. “Line 314 - typo "PPP6" for 'PP6" and second time "phosphatase" after brackets is obsolete.” We changed the name to PPP6C, the official name of protein phosphatase 6 catalytic subunit according to the HUGO Gene Nomenclature Committee (line 422).

  1. “Line 320 - typo "on mRNAs" might be replaced by "in mRNAs".”

We have changed this to “mRNA N6 -methyl-adenosine (m6 A) methylation” (line 430).

  1. “Line 340 - typo "phosphorylates" might be replaced by "phosphorylate".”

MAPK8 and JNK1 are the same molecule, so “phosphorylates” is correct, referring to the singular protein (line 448).

  1. “Lines 430 and 432 - sentence seems misleading and missing words or conjugation between parts.”

Thank you for pointing out the confusion. We rewrote the sentence as “In coordination with the ubiquitin-proteasome system, autophagy plays essential roles in sequestering oxidized proteins in the lysosome/vacuole for degradation to maintain homeostasis [153-155]. However, autophagy is also involved in oxidative stress-induced cell death [156]” (lines 544-546).

  1. “Line 432 - "there are abundant" should be replaced by "there is abundant".”

We changed “are” to “is” (line 549).

  1. “Line 478 - "redox regulated" might be replaced by "redox-regulated".”

In this case, “redox regulated” is not modifying a noun, so it is not correct to include a hyphen (line 610).

  1. “Line 486 - "stimulate" should be replaced by "stimulates".”

We meant to express “can stimulate” and we have added “can” to clarify this (line 624).

  1. “Lines 491-495 - the sentence is too ling and hard for understanding so I suggest to break it in 2 phrases.”

This is a good suggestion, and we have separated this into 2 sentences.

  1. “Lines 525-530 are more appropriate at the start of section rather that as a concluding part.”

We agree with the reviewer. We have moved this part of the text to the beginning of section 4.2.

  1. “Line 602 - comma after "on, this organelle" is obsolete.”

The reviewer is correct, and we deleted the comma (line 921).

Reviewer 3 Report

The manuscript “How cells deal with the fluctuating environment: Autophagy regulation under stress” is well written, easy to read.

There are some minor issues to be corrected.

In mammals, NFE2L2 is the gene, and if one refers to protein Nrf2 is the common name to use.

Nrf2 is the master antioxidative response transcription factor, but it can be regulated by other, non-oxidative-relate pathways.

Round 2

Reviewer 2 Report

Dear colleagues!
I appreciate the efforts made during manuscript revision and have no further queries.

Best, Reviewer.